# Time-dependent solid-state molecular motion and colour tuning of host-guest systems by organic solvents

Yu-Dong Yang [1], Xiaofan Ji[2], Zhi-Hao Lu [1], Jian Yang [1], Chao Gao [1], Haoke Zhang [2], Ben Zhong Tang [2]*, Jonathan L. Sessler [3,4]* & Han-Yuan Gong[1]*

Host-guest complex solid state molecular motion is a critical but underexplored phenomenon. In principle, it can be used to control molecular machines that function in the solid state. Here we describe a solid state system that operates on the basis of complexation between an all-hydrocarbon macrocycle, **$D_{4d}$-CDMB-8**, and perylene. Molecular motion in this solid state machine is induced by exposure to organic solvents or grinding and gives rise to different co-crystalline, mixed crystalline, or amorphous forms. Distinct time-dependent emissive responses are seen for different organic solvents as their respective vapours or when the solid forms are subject to grinding. This temporal feature allows the present **$D_{4d}$-CDMB-8**⊃perylene-based system to be used as a time-dependent, colour-based 4th dimension response element in pattern-based information codes. This work highlights how dynamic control over solid-state host-guest molecular motion may be used to induce a tuneable temporal response and provide materials with information storage capability.

[1] College of Chemistry, Beijing Normal University, Xinjiekouwaidajie 19, Beijing 100875, P. R. China. [2] Department of Chemistry, HKUST Jockey Club Institute for Advanced Study, Institute of Molecular Functional Materials, Division of Biomedical Engineering, State Key Laboratory of Molecular Neuroscience, Division of Life Science, The Hong Kong University of Science and Technology, Clear Water Bay, Kowloon, Hong Kong, China. [3] Department of Chemistry, The University of Texas at Austin, 105 East 24th Street, Stop A5300, Austin, TX 78712-1224, USA. [4] Department of Chemistry and Center for Supramolecular Chemistry and Catalysis, Shanghai University, 99 Shangda Road, Shanghai 200444, China. *email: tangbenz@ust.hk; sessler@cm.utexas.edu; hanyuangong@bnu.edu.cn

Solid-state molecular motion is a critical but underexplored phenomenon in nature. However, it is becoming appreciated as a powerful tool in crystal engineering that can be exploited to produce inter alia highly efficient catalysts[1–4], gas storage systems[5–8], and molecular machines (e.g., molecular switches[9–14], rotors[15–20], and shuttles[21]). The discovery of unusual solid-state molecular motion phenomena is of interest; along with accompanying dynamic regulation strategies such findings could allow for further advances in this area and might lead to the preparation of functional materials[22–29].

Molecular motion involving host-guest complexes[30–32] (e.g., inclusion complexes, pseudo rotaxanes) and mechanically interlocked molecules (MIMs)[33,34] (e.g., rotaxanes and catenanes) has been extensively explored to create molecular machines that operate under solution phase conditions[35]. In contrast to what is seen in solution, where molecular motion is typically facile, the close packing characteristic of most solid-state forms makes molecular movement and its controlled induction difficult. This has made it challenging to develop systems displaying motion in the solid state. Finding ways to regulate this motion has proved even more difficult. One approach could involve the use of liquid additives (e.g., solvents) that could serve as effective mobilizers to promote motion within the solid state. The viability of this strategy has been documented in the case of liquid crystals[36–38], classic industrial plastic processes[39–41], and several crystal-to-crystal transformations[42–45]. However, to our knowledge, solid-state molecular machines that rely on the solvent-based modification of host-guest complexes to achieve molecular motion have not been reported. Nor have time-dependent, motion-induced changes in the emissive features of solid materials been observed. To the extent such effects are demonstrated it could lead to an ostensibly new type of molecular machine where solvents provide the fuel to drive the system and motion is reflected in easy-to-discern changes in the solid-state properties. In the limit, such machines could provide dynamic constructs that allow information to be stored, manipulated, and read out in an input-specific, time-dependent manner.

Here we report a solid-state molecular machine based on an all-hydrocarbon host-guest construct derived from $D_{4d}$-CDMB-8 and perylene (Py). As-prepared crystalline forms of $D_{4d}$-CDMB-8 and Py react with a variety of solvent molecules (e.g., acetonitrile, THF, nitrobenzene, toluene, etc.) to produce several solid-state forms. These forms contain (1) a core host-guest complex stabilized by non-covalent π–π and CH–π interactions between the $D_{4d}$-CDMB-8 host and the Py guest and (2) a liquid domain arising from solvent capture that promotes molecular movement in the solid state. As detailed below, different solid-state forms, including, (i) co-crystalline phase, (ii) mixed crystalline, and (iii) amorphous state materials are obtained in the presence or absence of different solvents. These forms are characterized by different emissive features. This allows the various solvents to be distinguished. Other stimuli, such as mechanical impact (e.g., grinding), also lead to changes in the structural and optical properties. Moreover, a time-dependent response is seen. Systems derived from $D_{4d}$-CDMB-8 and Py thus act as rudimentary molecular machines where the inputs are solvents and the result of motion is a readily discernible structural and optical response (Fig. 1a). The time-dependent nature of the transformations, as well as its origins in the choice of solvent and solid-state forms, has allowed the $D_{4d}$-CDMB-8 and Py system to be elaborated to produce a set of 4D codes wherein time-dependent changes provide one information coding dimension and colours and patterns the other three (Fig. 1b).

## Results

### Host-guest system solid-state molecular motion determinants.
CDMB-8 is an all-hydrocarbon macrocycle that exists in the form

of two isomers, which are not easily interconverted[46]. In our initial study, we found that one isomer, namely $D_{4d}$-CDMB-8, could act as a good receptor for curved aromatic molecules (e.g., $C_{60}$ and $C_{70}$) both in solution and the solid state[46]. This finding led us to explore whether the isomeric species, $C_s$- and $D_{4d}$-CDMB-8, would interact with planar aromatic molecules. Perylene (Py) was chosen for initial studies. It was found that little if any discernible change in either the UV–Vis absorption, fluorescence emission or $^1$H NMR spectra was seen when Py was mixed with either $C_s$- or $D_{4d}$-CDMB (Supplementary Figs. 2–5). On the other hand allowing a mixture of $D_{4d}$-CDMB-8 containing 1 molar equiv. of Py in THF/CH$_3$CN (1/1, $v/v$) to undergo slow evaporation yielded green-yellow prismatic single crystals of [($D_{4d}$-CDMB-8)$_2$⊃(Py•6CH$_3$CN)•Py•2THF] (a solid-state construct referred to as $C_\alpha$, Fig. 2c). Evidence for the formation of a $D_{4d}$-CDMB-8⊃Py co-crystalline complex in $C_\alpha$ came from a single crystal X-ray diffraction analysis (Supplementary Figs. 6–8). Each periodic repeat unit contains two macrocycles and two perylenes, as well as six CH$_3$CN and two THF molecules.

Two interaction modes were observed for the perylene guests. One guest is seen to reside outside of $D_{4d}$-CDMB-8 being held there through presumed CH–π interactions. Another perylene molecule is located between two neighbouring macrocyclic cavities, being stabilized through possible CH-π and π-π interactions with $D_{4d}$-CDMB-8. An extended 1D linear packing structure is seen. An unusual cluster of six CH$_3$CN molecules serves as bridge between two inserted perylene moieties. Further study revealed that $C_\alpha$ could be easily prepared on a gram scale using the above approach (Supplementary Movie 1). A powder X-ray diffraction (PXRD) analysis of the resulting bulk sample proved in good agreement with the simulated PXRD pattern calculated using the single crystal data for $C_\alpha$ (Supplementary Fig. 9).

When $C_\alpha$ was allowed to stand in the air for six days or subject to vacuum for 5 h at ambient temperature, partial loss of solvent was seen. The resulting solid form, referred to as $I_\alpha$, is characterized by lattice features that are similar to those seen in $C_\alpha$ (Fig. 2a and Supplementary Figs. 16–19). In order to remove all the organic solvent molecules present in the complex between $D_{4d}$-CDMB-8 and Py, $C_\alpha$ was subject to grinding using a mortar and pestle. This action produced an amorphous material ($A_m$, Fig. 2b), as inferred from a PXRD analysis (Supplementary Fig. 16b). A $^1$H NMR spectral study revealed that $A_m$ contains only macrocycle and perylene and is free of residual organic species (Supplementary Fig. 20). The same $A_m$ product can also be generated easily by simply grinding a 1:1 molar ratio mixture of $D_{4d}$-CDMB-8 and Py in the absence of a solvent. Treating $A_m$ with THF/CH$_3$CN (1/1, $v/v$), either by mixing with the solvents directly or exposing the solid material to their vapours (assumed to be saturated in air at 298 K), served to regenerate $C_\alpha$ (Supplementary Fig. 16c).

The weak interactions within $C_\alpha$ as inferred from the initial single crystal X-ray diffraction studies led us to hypothesize that the above liquid and organic vapour-induced phenomenological changes were structural in origin. In an initial effort to test this hypothesis, studies were carried out using nitrobenzene (NB) vapour. In fact, a particularly complex response was seen. A combination of $^1$H NMR spectral, in situ single crystal X-ray diffraction, and PXRD analyses (Supplementary Figs. 23 and 26) leads us to suggest that a stable intermediate material ($I_\beta$) is produced as the result of nitrobenzene vapour covering the surface of $C_\alpha$ in 20 min (Fig. 2d). In this case, little apparent nitrobenzene-induced molecular motion is observed, and the lattice parameters associated with $D_{4d}$-CDMB-8 and Py are retained.

Over longer time scales (i.e., from 3 to 21 h post exposure), a crystalline transformation is seen. Structural analyses provided

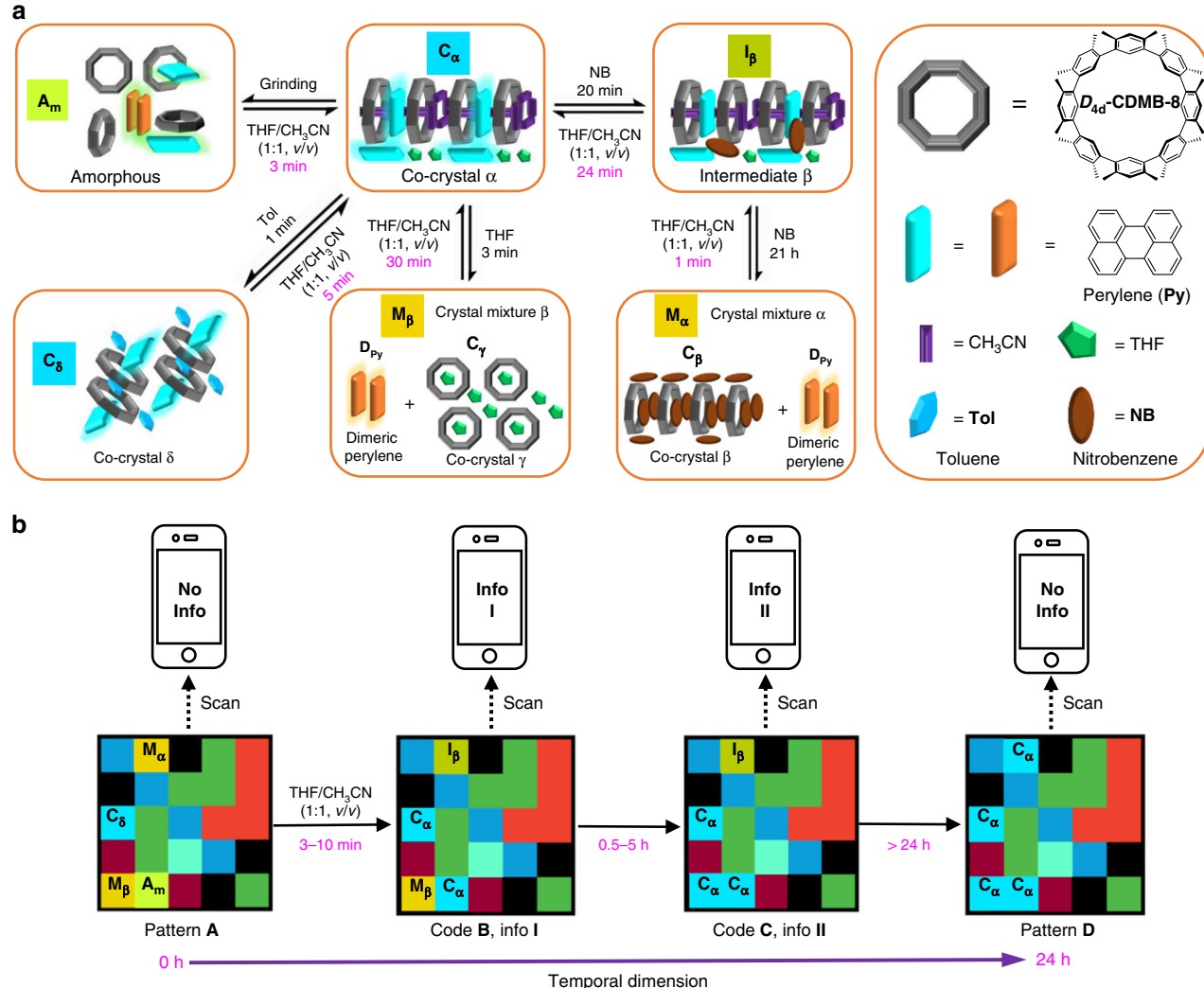

**Fig. 1 Schematic representations of solvent promoted solid-state molecular motion involving host-guest complexes and the 4D information coding the accompanying changes permit. a** An organic solvent, applied to the solid forms as either a liquid or vapour, promotes solid-state molecular motion and leads to transformation between co-crystalline phases, as well as production of mixed crystalline states and an amorphous form. Also shown by use of highlighting (light blue vs orange) is the time resolved colour response produced under both room light and UV illumination (365 nm). See text for further details. **b** 4D information coding via a solvent-induced time-dependent vapoluminescent response.

support for the notion that all the components within $I_\beta$ undergo nitrobenzene vapour-induced motion to produce $M_\alpha$, a mixed crystalline solid material containing separate crystals of both dimeric perylene ($D_{Py}$) and [$D_{4d}$-**CDMB-8**⊃(NB)₂•2NB] ($C_\beta$) (Fig. 2e, Supplementary Figs. 23 and 27). Based on this observation, we infer that nitrobenzene promotes molecular motion in the solid state by providing liquid domains within the crystals of $C_\alpha$.

Support for the above suggestion came from the finding that when crystals of $C_\alpha$ were recrystallized from nitrobenzene, product $M_\alpha$ was also obtained. The mixed crystalline material, $M_\alpha$, could be transformed back to $I_\beta$ and $C_\alpha$, albeit with different dynamics (1 min vs. 24 h, respectively), via exposure to THF/CH₃CN (1/1, v/v) vapour (Supplementary Fig. 25). Dissolution and recrystallization of $M_\alpha$ in THF/CH₃CN (1/1, v/v) also allowed for recovery of $C_\alpha$. To our knowledge, this solvent-induced transformation between a co-crystalline species containing a host-guest complex and a corresponding crystalline mixture containing separated host and guest species is without precedent in the literature. It represents what to our knowledge is a unique type of

solid-state molecular motion. Thus, efforts were made to explore it further.

As a first step, we sought to explore the effect of different organic solvents. When $C_\alpha$ was exposed to THF vapour, a combination of ¹H NMR spectral, PXRD, and X-ray diffraction analyses (Supplementary Figs. 31–33) provided support for the conclusion that a new crystalline mixture, referred to as $M_\beta$, containing crystals of both [$D_{4d}$-**CDMB-8**⊃THF•THF] ($C_\gamma$) and $D_{Py}$, was produced (Fig. 2g). When $C_\alpha$ was dissolved in THF and subject to slow evaporation, only blocks of colourless $C_\gamma$ and yellow $D_{Py}$ crystals were obtained as deduced from a single crystal diffraction analysis. On this basis, we propose that the co-crystalline species $C_\alpha$ undergoes disassembly upon exposure to THF, which then allows formation of the mixed crystalline material, $M_\beta$. Under identical exposure conditions THF vapour promotes this conversion much faster (3 min) than nitrobenzene vapour (21 h). Again, apparent molecular motion underlies the observed solid-state structural switching (Supplementary Fig. 32).

Exposure of the co-crystalline form $C_\alpha$ to toluene (Tol), either in liquid or vapour form, led to the formation of another

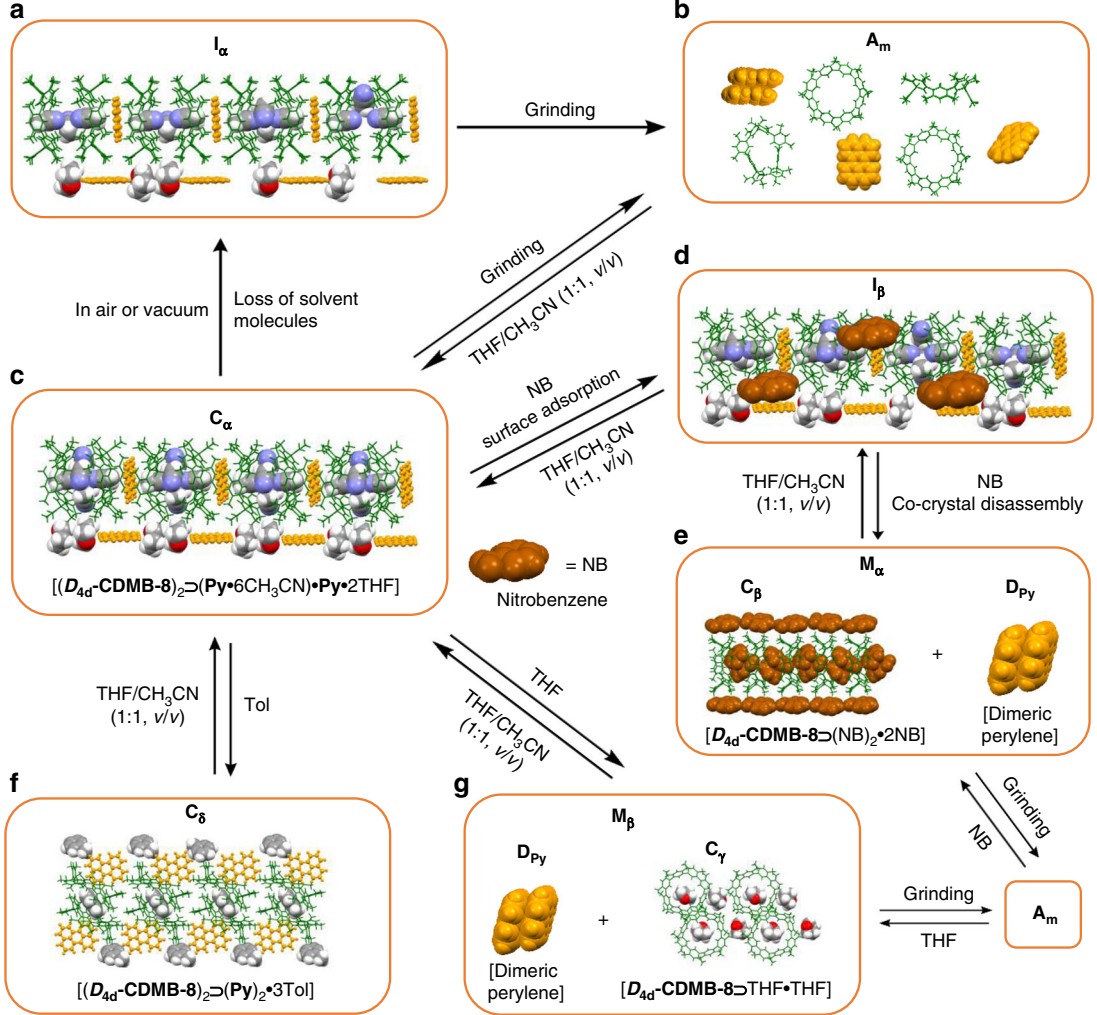

**Fig. 2 Complexation and decomposition of host-guest complexes involving solid-state molecular motion.** Structures of, and transformations between, the co-crystalline material $C_\alpha$, mixed crystalline species (e.g., $M_\alpha$ and $M_\beta$), co-crystalline materials (i.e., $I_\alpha$, $I_\beta$, and $C_\delta$), and the amorphous form ($A_m$) produced by exposure to organic solvents (vapour or liquid forms) or other treatments (e.g., grinding). **a, b, d, e, g** Suggested structures of $I_\alpha$, $A_m$, $I_\beta$, $M_\alpha$, and $M_\beta$ base on PXRD analysis, [1]H NMR, and fluorescence spectroscopic studies. **c, f** Single crystal structure of co-crystals $C_\alpha$ and $C_\delta$.

co-crystalline solid (referred to as $C_\delta$; Fig. 2f). Based on a qualitative comparison of the structures involved, considerable molecular motion is associated with this transformation. The PXRD pattern of $C_\delta$ proved to be a good match with the simulated pattern for a new co-crystalline material [($D_{4d}$-**CDMB**-8)$_2$⊃(**Py**)$_2$•3Tol] (Supplementary Fig. 36b). [1]H NMR spectral studies of $C_\delta$ in CDCl$_3$ exposed to toluene vapour provided support for the notion that 3 molar equiv. of toluene replace all the THF and CH$_3$CN molecules originally present in $C_\alpha$ thus producing $C_\delta$ (Supplementary Fig. 36c). When $C_\alpha$ was dissolved in toluene and subject to slow evaporation, only green-yellow blocks of [($D_{4d}$-**CDMB**-8)$_2$⊃(**Py**)$_2$•3Tol] crystals were obtained as deduced from single crystal diffraction analyses. A single crystal X-ray diffraction structure of this material (i.e., [($D_{4d}$-**CDMB**-8)$_2$⊃(**Py**)$_2$•3Tol]) revealed that the perylene guest is inserted head on into the cavity of the $D_{4d}$-**CDMB**-8 host and that one toluene molecule is located between two $D_{4d}$-**CDMB**-8 macrocycles (Supplementary Figs. 12, 13). On this basis, we propose that the $C_\alpha$ co-crystalline material undergoes solid-state molecular motion upon exposure to toluene vapour and then forms a new co-crystalline material, $C_\delta$. This process is reversible. Exposing $C_\delta$ to saturated THF/CH$_3$CN (1/1, *v/v*) vapour served to regenerate $C_\alpha$ (Supplementary Fig. 37). Under conditions identical to those used

in the case of nitrobenzene and THF, the toluene vapour-induced conversion between $C_\alpha$ and $C_\delta$ proved relatively fast (1–5 min).

**Emission-based colour response.** A change in the fluorescence colour from green to orange was seen when $C_\alpha$ was converted to a mixed crystalline form (e.g., $M_\alpha$ or $M_\beta$) by exposure to various organic solvents (either as vapours or used as for recrystallization) (cf. Supplementary Movie 2, and Supplementary Figs. 28, 34, and 46). The corresponding emission band appears at around 590 nm under conditions of UV illumination ($\lambda_{ex}$ = 365 nm). This matches the emission produced by the presumably aggregated **Py** in crystalline $D_{Py}$. This correspondence was taken as evidence that mixed crystalline solids, such as $M_\alpha$ or $M_\beta$, contain $D_{Py}$ crystalline domains and that these are responsible for the observed emission features. $I_\alpha$ and amorphous material ($A_m$) was characterized by a green-yellow emission around 500 nm (Fig. 3, Supplementary Fig. 22). Crystals $C_\alpha$ and $C_\delta$ gave rise to a green luminescence ($\lambda_{em}$ = ca. 480 nm) when subject to excitation at 365 nm using a handheld ultraviolet lamp. This emission is similar to that produced by perylene in solution at high concentrations (e.g., 10 mM in THF/CH$_3$CN (1/1, *v/v*)) (Supplementary Fig. 21). This similarity leads us to propose that in the

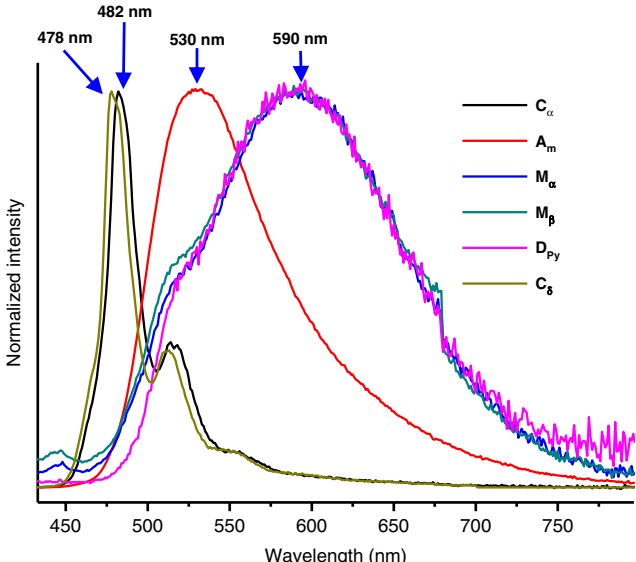

**Fig. 3 Normalized emission spectra of various solid forms considered in this study.** The spectra of $C_\alpha$, $A_m$, $M_\alpha$, $M_\beta$, $D_{Py}$, and $C_\delta$ are shown ($\lambda_{em}$ = 365 nm).

**Table 1 Photophysical properties of various solid materials.**

|  | Photos $\lambda_{em}$ = 365 nm | $\lambda_{em}$ (nm) | $\Phi_f$ |
|---|---|---|---|
| $C_\alpha$ |  | 482, 515, 560 | 0.41 |
| $A_m$ |  | 530 | 0.45 |
| $M_\alpha$ |  | 590 | 0.07 |
| $M_\beta$ |  | 590 | 0.12 |
| $D_{Py}$ |  | 590 | 0.12 |
| $C_\delta$ |  | 478, 513, 550 | 0.48 |

co-crystalline solid states, $D_{4d}$-CDMB-8 serves to disperse the perylene molecules thus maintaining them in monomeric form. The fluorescence emission peak ($\lambda_{em}$), quantum yields ($\Phi_f$), and fluorescent lifetimes ($\tau_f$) of $C_\alpha$, $A_m$, $M_\alpha$, $M_\beta$, $D_{Py}$, and $C_\delta$, are summarized (see Table 1, and Supplementary Figs. 44, 45).

Given the clear distinction in the luminescence features of the different solid forms, the emission colours and spectra could be used to follow the structural changes associated with the conversion between the co-crystalline and mixed crystalline forms. Thus, these spectral changes could be used to monitor exposure to organic solvents. For instance, in a reflection of the complex time-dependent structural changes produced when $C_\alpha$ was exposed to nitrobenzene vapour, the emission of $C_\alpha$ at 482 nm was found to be quenched initially (a finding correlated with the proposed initial formation of $I_\beta$), followed by the production of an aggregation emission of **Py** ascribed to $D_{Py}$ as complete conversion from $C_\alpha$ to $M_\alpha$ occurs (Supplementary Figs. 28 and 30).

**Dynamic control over molecular motion and associated colour tuning.** As highlighted by the conversion from $C_\alpha$ to $M_\alpha$ induced by nitrobenzene, the organic vapour-induced changes in the luminescent features of the various solid forms generated from $D_{4d}$-CDMB-8 and **Py** proved time-dependent. The temporal response was found to be a function of both the species in question and the organic solvent employed (Fig. 4). For instance, the conversion of $C_\alpha$ to $C_\delta$, $M_\beta$, or $I_\beta$ and then to $M_\alpha$ required 1, 3, and 20 min followed by 21 h upon exposure to toluene, THF, and nitrobenzene vapour, respectively (Fig. 4a). Likewise, the regeneration of $C_\alpha$ from $A_m$, $C_\delta$, $M_\beta$, or $M_\alpha$ via treatment with an identical THF/CH$_3$CN (1/1, v/v) vapour mixture, required 3 min, 5 min, 30 min, and 24 h, respectively (Fig. 4b). Further inter-species transformations are summarized in Supplementary Fig. 41.

The solvent-based differentiation is ascribed to relative affinities for the $D_{4d}$-CDMB-8 host, as compared to **Py**, in the solid state. When the interaction between $D_{4d}$-CDMB-8 and the organic solvent is strong, as true in the case of, e.g., THF and nitrobenzene, decomposition of the co-crystalline species comprising **Py** and $D_{4d}$-CDMB-8 occurs. The net result is formation

of mixed crystalline materials containing $D_{4d}$-CDMB-8⊃solvent and $D_{Py}$. Such solid forms are characterized by an orange $D_{Py}$ emission ($\lambda_{em}$ = 590 nm). In contrast, treatment with an organic species (e.g., toluene) with weaker affinity for $D_{4d}$-CDMB-8 (compared with **Py**) only gives rise to a modified co-crystalline $D_{4d}$-CDMB-8⊃**Py**•solvent species (solvent = toluene). Such forms maintain the monomeric **Py** blue emission ($\lambda_{em}$ = 480 nm).

**4D code system.** The structural changes and associated diagnostic luminescent features produced via the various solid forms upon exposure to organic solvent vapours were found to proceed on different, but highly reproducible time scales (Supplementary Figs. 41–46). These differences and the temporal fidelity associated with the organic vapour-induced interconversions between $C_\alpha$, $A_m$, $I_\beta$, $M_\alpha$, $M_\beta$, and $C_\delta$, led us to consider that solid forms generated from $D_{4d}$-CDMB-8 and **Py** could be used to create a time-dependent dynamic 4D code system. As an initial test of this proposition, $A_m$ was loaded onto scraps of paper (0.5 × 0.5 cm) to provide a first fluorescent block. Independent treatment with nitrobenzene, THF, and toluene vapour was then used to generate another three fluorescent blocks, namely $M_\alpha$, $M_\beta$, and $C_\delta$ (Fig. 5a). A printed colour pattern $A_0$ was then generated by means of a commercial colour printer (Fig. 5b). Fluorescent blocks containing $A_m$, $M_\alpha$, $M_\beta$, and $C_\delta$ were then added into the printed pattern to obtain a 3D colour-based pattern ($A_1$). Pattern $A_1$ was found to produce fluorescent pattern **A** when exposed to UV light at 365 nm using an ultraviolet lamp under otherwise normal laboratory conditions (Supplementary Movie 3). Exposing pattern **A** to THF/CH$_3$CN (1/1, v/v) vapour produces patterns **B-D** over different time scales, namely 3–10 min, 0.5–5 h, and greater than 24 h, respectively (cf., Supplementary Movie 4).

To demonstrate the power of this time-dependent approach to coding, we programmed information sets **I** and **II** within the time-dependent patterns **B** and **C**, respectively. The information in question could be accessed directly by scanning these two codes by means of a smart phone. However, different delay times, namely 3–10 min and 0.5–5 h, were required to read out the information (Supplementary Movie 4). As the result of this time-dependent feature, a 3D code system (a planar coloured array) is transformed into a dynamic information coding system characterized by 4D complexity. The net result is a set of code systems with a high degree of inherent confidentiality. This reflects the fact that (i) the different code arrangements (placement of coloured building blocks), (ii) the light source (UV vs Vis), (iii) choice of organic vapour, and (iv) the exposure and monitoring time scales may be used to generate various codes that allow information to be read out in a temporally defined manner.

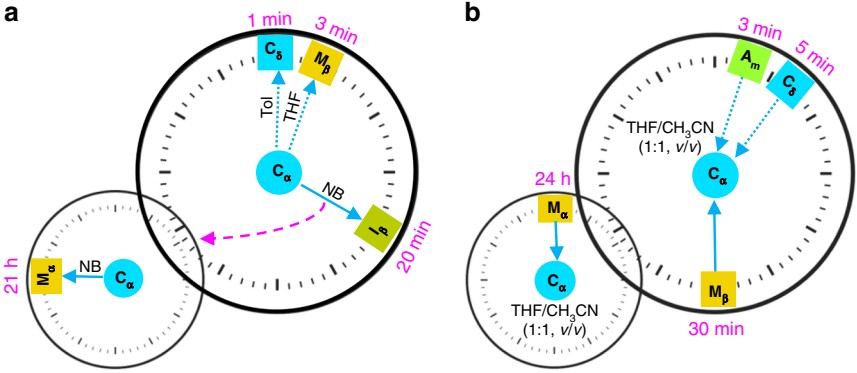

**Fig. 4 Time-dependent transformation between various solid materials induced by organic solvents. a** Time-dependent conversion of $C_\alpha$ to $C_\delta$, $M_\beta$, or $I_\beta$ and then to $M_\alpha$ seen upon exposure to different organic vapours. **b** Time-dependence of the regeneration of $C_\alpha$ from different solid forms, namely $A_m$, $C_\delta$, $M_\beta$, and $M_\alpha$, triggered by exposure to the same THF/CH$_3$CN (1/1, $v/v$) vapour mixture.

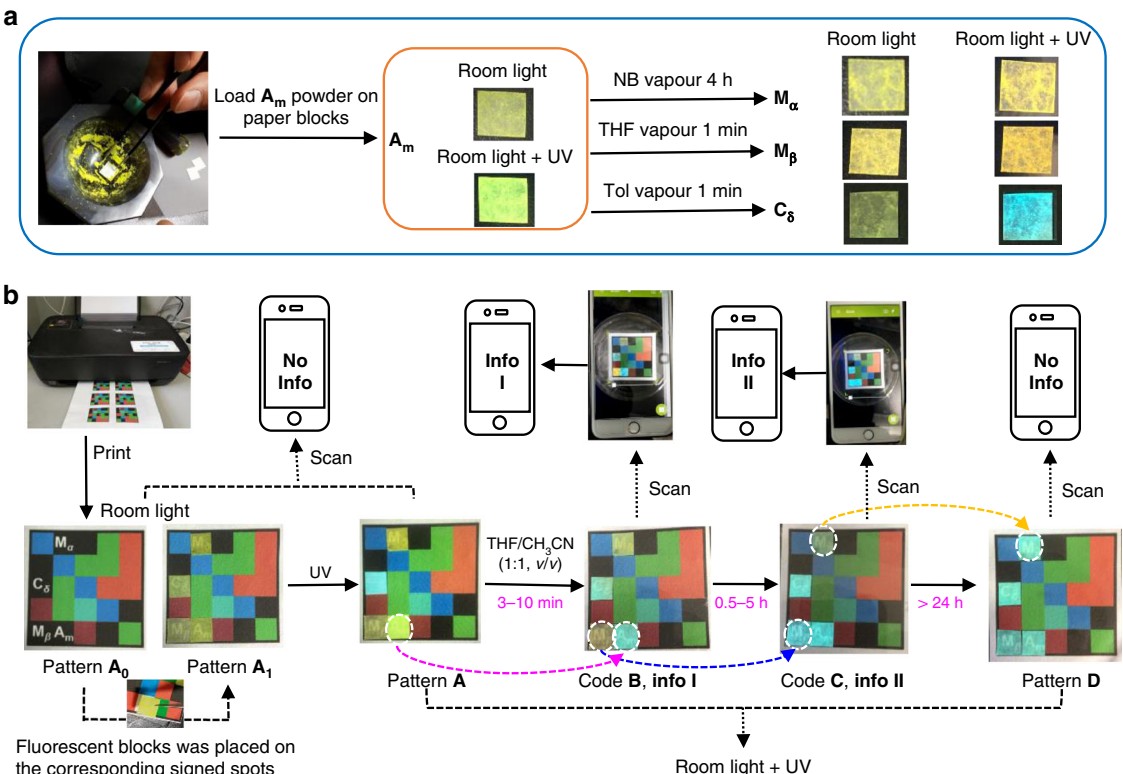

**Fig. 5 Schematic representation of a 4D code information system based on a time-dependent organic vapour-based response. a** Preparation of code blocks $A_m$, $M_\alpha$, $M_\beta$, and $C_\delta$. **b** Transformation and masking mad possible by using (i) an original printed colour pattern, (ii) room and/or UV light, and (iii) the time-dependent organic vapour-induced changes in the luminescent features of the constituent code blocks, $A_m$, $M_\alpha$, $M_\beta$, and $C_\delta$. Additional photographs of the various patterns are included in the Supporting Information.

Treating the original pattern **A** with other vapours or set in air was found to give rise to other patterns (**E**–**J**) characterized by different formation lifetimes (see Supplementary Figs. 47–49).

## Discussion

We have shown that molecular motion can be induced by exposure of solid-state host-guest complexes to appropriately chosen organic solvents and that the resulting changes in structure give rise to distinct luminescent features. In the case of the present materials, which are based on the all-hydrocarbon components $D_{4d}$-**CDMB-8** and **Py**, the underlying molecular scale

motion and the associated changes in the solid-state structures was inferred from a combination of single crystal X-ray diffraction analyses and PXRD studies. A tuneable temporal response, which proved to be a function of both the solid form employed and the chosen organic solvent, was seen. This allowed a dynamic 4D code library to be developed wherein multiple independent keys, including time and a specific organic vapour, are needed in order to read out pre-programmed information.

## Methods

**General considerations**. Deuterated solvents were purchased from Cambridge Isotope Laboratory (Andover, MA). All other solvents and reagents were purchased

commercially (Aldrich, Acros, or Fisher) and used without further purification. NMR spectra were recorded on Bruker AVANCE III 500WB, AVANCE 400, or 400JNM-ECZ400S spectrometers. The [1]H NMR chemical shifts are referenced to residual solvent signals (tetrachloroethane-$d_2$ (TCE-$d_2$): $\delta_H$ = 5.95 ppm, $\delta_C$ = 74.10 ppm. CDCl$_3$: $\delta_H$ = 7.26 ppm. THF-$d_8$: $\delta_H$ = 1.75, 3.60 ppm. CH$_3$CN-$d_3$: $\delta_H$ = 1.94 ppm). UV–vis spectra were collected on a Shimadzu UV-2450 instrument. Fluorescence emission spectra and lifetimes ($\tau_f$) were collected on an Edinburgh Instruments FS5 spectrometer. Fluorescence quantum yields ($\Phi_f$) were obtained using a HAMAMATSU Quantaurus-QY instrument. PXRD studies were carried out using a Shimadzu XRD-7000 setup. Pictures and movies were recorded on a smart phone (Vivo X23) or an industrial digital camera (E3ISPM05000KPA) linked to a stereo microscope (Carton SPZT-50PFM). The information codes were registered on www.colorzip.com and read by a smart phones (iPhone 5S) using the COLORCODE$^{TM}$ app, which at the time the work was performed could be downloaded for free from the Apple app store.

**Single crystal X-ray diffractions**. Unless otherwise noted, single crystals used to obtain the X-ray diffraction structures reported in the main text grew as yellow prisms, blocks, or colourless prisms. The data crystals used for single crystal analyses were cut from clusters of the corresponding crystals. The data were collected on Saturn724 + (2 × 2 bin mode) or SuperNova, Dual, Cu at Home/Near, AtlasS2 diffractometers. Data reduction was performed using the CrystalClear (Rigaku Inc., 2007) or CrysAlisPro 1.171.39.32a (Rigaku OD, 2017) software packages. The structures were refined by full–matrix least–squares on F$^2$ with anisotropic displacement parameters for the non–H atoms using SHELXL-2014[47]. The hydrogen atoms were calculated in idealized positions with isotropic displacement parameters set to 1.2 × Ueq of the attached atom (1.5 × Ueq for methyl hydrogen atoms). Definitions used for calculating R(F), Rw(F2) and the goodness of fit, S, are given in Supplementary Tables 1, 2. Neutral atom scattering factors and values used to calculate the linear absorption coefficient are from the International Tables for X-ray Crystallography (1992)[48]. All ellipsoid figures were generated using SHELXTL/PC[49].

**Synthesis of CDMB-8**. Under an argon atmosphere, a mixture of **1** (600 mg, 0.88 mmol), **2** (500 mg, 0.88 mmol), Pd(dppf)$_2$Cl$_2$·CH$_2$Cl$_2$ (148 mg, 0.18 mmol), Cs$_2$CO$_3$ (14.4 g, 44 mmol), and 1000 mL deaerated toluene were added to a 2 L three-necked round-bottomed flask. The reaction mixture was heated under reflux for 12 h. After allowing to cool to room temperature, the volatiles were removed using a rotary evaporator. The resulting residue was dissolved in 100 mL CH$_2$Cl$_2$, then filtered through a short neutral alumina column and washed with CH$_2$Cl$_2$ (50 mL). The volatiles (primarily CH$_2$Cl$_2$) were removed from the filtrate via rotary evaporation. The resulting residue was dissolved in 50 mL cyclohexane and purified by a neutral alumina column using cyclohexane as the eluent to give analytically pure **$C_s$-CDMB-8** and **$D_{4d}$-CDMB-8** as a white solid 110 mg (15%) and 180 mg (25%), respectively. **$C_s$-CDMB-8**: [1]H NMR (500 MHz, TCE-$d_2$, 278 K) $\delta$ (ppm): 7.08 (s, 2H), 7.04 (s, 2H), 7.02 (s, 2H), 7.01 (s, 1H), 6.96 (s, 1H), 6.91 (s, 1H), 6.89 (s, 2H), 6.86 (s, 2H), 6.83 (s, 2H), 6.81 (s, 2H), 2.16 (s, 6H), 2.12(s, 6H), 2.11 (s, 6H), 2.09 (s, 6H), 2.08 (s, 6H), 2.07 (s, 12H), 2.02 (s, 6H). **$D_{4d}$-CDMB-8**: [1]H NMR (500 MHz, TCE-$d_2$, 278 K) $\delta$ (ppm): 7.03 (s, 8H), 6.75 (s, 8H), 2.01 (s, 48H). Both products were further characterized by single crystal X-ray diffraction analysis (Supplementary Fig. 1).

**Preparation of co-crystal materials**. Subjecting mixtures of $D_{4d}$-CDMB-8 (1.00 mM), and 1 molar equiv. of perylene in THF/CH$_3$CN (1/1, $v/v$) or toluene (Tol) to slow evaporation resulted in the formation of single crystals of [($D_{4d}$-CDMB-8)$_2$⊃(Py•6CH$_3$CN)•Py•2THF] ($C_\alpha$) or [($D_{4d}$-CDMB-8)$_2$⊃(Py)$_2$•3Tol] ($C_\delta$), respectively. Separately, single crystals of [$D_{4d}$-CDMB-8⊃(NB)$_2$•2NB] ($C_\beta$) were obtained via the slow evaporation of $D_{4d}$-CDMB-8 (1.00 mM) in nitrobenzene (NB), or by dissolving $C_\alpha$ in nitrobenzene and subjecting the resulting solution to slow evaporation. These various single crystals were analyzed by X-ray diffraction methods.

**Organic solvent vapour treatment equipment and conditions**. Small borosilicate glass fragments (thickness: 0.3 cm) were placed in a 100 mL petri dish, and a borosilicate glass plate (4 × 4 × 0.3 cm) was stacked up on these borosilicate glass fragments. The test materials were loaded on a small borosilicate glass block (0.5 × 0.5 × 0.1 cm) and set in the centre of the glass plate. Two millilitres of organic solvents liquid were dropwise added to the bottom of the petri dish, which was immediately covered with the lid (Supplementary Fig. 14). Note: These conditions provide the organic solvent vapours in near-saturated form in air at room temperature (298 K).

**In situ time-dependent emission spectra collection**. Briefly, the test material is loaded on a paper square (0.5 × 0.5 × 0.1 cm). The paper square is fixed on a quartz plate (3 × 1 × 0.1 cm) by means of a copper wire. The quartz plate is then placed in a 20 mL borosilicate glass bottle and set on the sample stage of an Edinburgh Instruments FS5 instrument. The excitation light (365 nm) is then focused on the sample and the spectra features recorded. At this juncture, 0.5 mL of the organic solvent in question was dropped in liquid form onto the bottom of the glass bottle

with an injector (Supplementary Fig. 15). The setup was then covered with the lid immediately (without tightening so as to allow venting to the atmosphere. these conditions provide the organic solvent vapours that are essentially saturated in air at room temperature (298 K)). The emission spectra were recorded as a function of time after the organic solvent in question was added.

**The equipment and conditions for 4D code generated**. A small glass fragment (thickness: 0.3 cm) was placed in a 100 mL petri dish, and a borosilicate glass plate (4 × 4 × 0.3 cm) was stacked up on theses glass fragments. Predetermined patterns were set on the glass plate, 2 mL of the organic solvent in question (in liquid form) was are added to the bottom of petri dish, which was immediately covered with the lid (these conditions provide the organic solvent vapours in near saturated form in air at room temperature (298 K)) (Supplementary Fig. 50). Pictures and movies were recorded under natural with/without UV light (using a commercial ultraviolet lamp (365 nm)). Movies showing the conversion of pattern **A$_0$** to pattern **D** upon treatment with THF/CH$_3$CN (1/1, $v/v$) vapour are given in Supplementary Movies 3, 4.

## Data availability
The X-ray crystallographic coordinates for structures reported in this study have been deposited at the Cambridge Crystallographic Data Centre (CCDC), under deposition numbers 1859991, 1859999, 1937315, 1937316, and 1937317. These data can be obtained free of charge from The Cambridge Crystallographic Data Centre via www.ccdc.cam.ac. uk/data_request/cif. And all other data supporting the findings of this study are available from the article and its Supplementary Information files or available from the corresponding authors upon reasonable request.

## Code availability
The scan software used in the present study is COLORCODE® (http://colorzip.com). At the time of publication this information reading application was available free of charge for use in code scanning, but not to the authors' knowledge for any other purpose.

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

## Acknowledgements

H.-Y.G. is grateful to the National Natural Science Foundation of China (21971022, 21472014, and 21672025), National Basic Research Program of China (973 Program 2015CB856502), the Young One Thousand-Talents Scheme, the Fundamental Research Funds for the Central Universities, the Beijing Municipal Commission of Education, the Beijing National Laboratory for Molecular Science (BNLMS), and Beijing Normal University for financial support. Partial support from the U.S. National Science Foundation (grant CHE-1807152 to J.L.S.), Shanghai University (to J.L.S.) and the Robert A. Welch Foundation (F-0018 to J.L.S.) is also acknowledged. B.Z.T. acknowledges the financial support from the National Science Foundation of China (21788102, 21490570 and 21490574), the Research Grant Council of Hong Kong (16308116 and C6009-17G), the Science and Technology Plan of Shenzhen (JCYJ20160229205601482, JCY20170307173739739, and JCYJ20170818113602462), and the Innovation and Technology Commission (ITC-CNERC149C01).

## Author contributions

Y.-D.Y. contributed to the experimental work, and wrote up the initial manuscript guided by H.-Y.G., X.J. and J.L.S. X.J. also helped for the design of information code. Z.-H.L., J.Y. and C.G. contributed to the synthesis of raw materials. H.Z. helped the solid-state fluorescence analysis. B.Z.T., J.L.S. and H.-Y.G. design this project and contributed to the manuscript writing. H.-Y.G. and Y.-D.Y. were responsible for the initial discoveries and oversaw the crystallographic studies.

## Competing interests

The authors declare no competing interests.
