## [Peer Review File · Nature Communications]

Reviewers' comments:

Reviewer #1 (Remarks to the Author):

In this manuscript, the authors represent a solid-state molecular motion system based on host-guest complex derived from D4d-CDMB-8 and perylene (Py). The crystalline solid Ca, which obtained from THF/CH₃CN (1/1, v/v) solution of D4d-CDMB-8 and Py, exhibited abundant structural transformations responding to various solvents with time-dependent features. After grinding or exposure to different solvent vapors, Ca transformed to multiple types of solid-state forms (co-crystalline phase, mixed crystalline phase or amorphous state). Due to the diverse aggregation modes of Py, these crystalline materials showed different time-dependent fluorescent changes. This kind of tunable temporal structural transformation and fluorochromism were utilized by authors to produce 4D codes for potential application of information storage and encryption. This work well used co-crystalline host-guest complexes to neatly regulate the structures and optical properties of solid-state materials. The designs of experiments were sufficient and clear and the results were well analyzed and displayed. I think this work can fit the requirements of Nature Communications and will have certain influence in several fields. Below are several minor issues for the authors to consider for the revision.

1. To obtain the original crystalline solid Ca, why did authors choose THF/CH₃CN (1/1, v/v) as solvent system? Will other solvent systems produce another solid-state form which also have abilities of tunable structural transformation and fluorochromism?
2. In Supplementary Figure 16 (a), how was the crystal structure of Ia confirmed? If it was analyzed by single crystal XRD experiment, the relevant data should be provided, if not, it should be explained in the annotation. According to the PXRD results, the crystal structures of Ca and Ia were not very consistent, thus it may not just be Ca minus several solvent molecules.
3. In Supplementary Figure 17, the PXRD patterns of Ca upon letting stand in air for 6 d and subjecting to vacuum for 5 h seem different to some extent, so I think these two solid-state forms could be respectively named as Ia1 and Ia2.
4. For the part of NB induced transformation, the intermediate crystalline solid I β was described as the results of NB vapor covering the surface of Ca. As shown in PXRD patterns, the crystal structure of Ca well remained after adsorption of NB for 20 min. Why not choose an intermediate state of structural transformation (it may be at about 1h according to time-dependent emission spectra of Ca to Ma) where the PXRD started to change?
5. In page 10, the first paragraph: "This matches the AIE emission produced by crystalline DPy." Perylene doesn't have AIE properties. The emission at 590nm was from aggregates of Py, which was more red-shifted and had much lower quantum yields compared to monomers and excimers. This was a classical manifestation of the ACQ phenomenon.
6. In Supplementary Figure 44 (a), the first line of the form which referred to the nonluminous crystalline materials should be "C β or C γ ".

Reviewer #2 (Remarks to the Author):

An exciting molecular motion in solid states was firstly found by treating with various organic solvents, and resulted different co-crystalline, mixed crystalline, or amorphous forms relying on the affinity of organic solvents with D4d-CDMB-8. Not only the liquid solvent, but also the solvent vapors could promote the transformation process. They exhibit different luminescence features as the solid states change, which give naked eye discrimination under room and/or UV light. The authors then created a time-dependent dynamic 4D code system. The information was hidden in initial until exposure to THF/CH₃CN vapor. Then multi information expressed with the passage of time. At last, the information was hidden again to prevent information leakage. Those phenomena are quite rare and interesting. I think it could be considered to publish on Nat. Comm. after addressing the following minor points.

1. The volume ratio of solvents THF/CH₃CN was 1/1, and the vapor was used assumed to be saturated in air at 298 K. So, did the author calculate the concentration ratio of the saturated vapor? If the ratio differs from that of the liquid solvents, which one is better for the molecular motion?

2. Could the transform procedure accelerate by raising the temperature when induced by solvent vapors?

3. The directly transformation between C δ and Am, C δ and M β , C δ and Ma, Ma and M β is not mentioned in the manuscript. Could they happen without state Ca?

4. Some English expressions need to be corrected:

The fifth line of the second paragraph: 'could serves' should be 'could serve';

The penultimate line of the sixteenth paragraph: 'a ultraviolet lamp' should be 'an ultraviolet lamp'.

Point-by-point response to Reviewers' concerns and queries:

Reviewer #1 (Remarks to the Author):

Comment 1. To obtain the original crystalline solid $C\alpha$, why did authors choose THF/CH₃CN (1/1, v/v) as solvent system? Will other solvent systems produce another solid-state form which also have abilities of tunable structural transformation and fluorochromism?

Answer: We thank the reviewer for these good questions and suggestion. Actually we also obtained some other host-guest co-crystals of macrocycle ***D*_{4d}-CDMB-8** and perylene with other solvent systems. One of them, [***D*_{4d}-CDMB-8**]₂⊃(Py)₂•3Tol (***C*_δ**) was detailed in the manuscript and ESI (Fig. 2 and Supplementary Figs. 12-13). Co-crystal ***C*_δ** was obtained from toluene (Tol) solution, which serves as an experimental demonstration that it is possible to tune the structural transformations and associated fluorochromism (Supplementary Figs. 36-43). Furthermore, a series of other solvent vapours (e.g., chlorobenzene, DCM, anisole, dioxane, bromobenzene, trimethylamine, and *m*-dibromobenzene) were found to promote the transformation from an amorphous solid form (***A*_m**) to co-crystalline species as shown in Supplementary Figure 46J-O (reproduced below for convenience). Obtaining diffraction grade crystals of these products has proved challenging. However, in our hands, THF/CH₃CN (1/1, v/v) yields co-crystalline solids of high quality. It is the reason why we choose to focus on this solvent system for this initial report. We do hope to follow up this *Nature Communication* (if accepted) with a full paper in due course. By then, we expect we will have expanded the lexicon of structurally characterized species.

Supplementary Figure 46. Time-dependent emission spectra for A_m recorded in the presence of various organic solvent vapours ($\lambda_{\text{ex}} = 365$ nm, voltage = 400 V, entrance slit width = 1 nm, exit slit width = 1 nm).

Comment 2. In Supplementary Figure 16 (a), how was the crystal structure of I_a confirmed? If it was analyzed by single crystal XRD experiment, the relevant data should be provided, if not, it should be explained in the annotation. According to the PXRD results, the crystal structures of C_a and I_a were not very consistent, thus it may not just be C_a minus several solvent molecules.

Answer: First, we would like to thank the reviewer for examining our data with such care. We agree with reviewer's consideration that according to the PXRD results, the crystal structures of C_a and I_a are not very consistent. However the characteristic peaks between C_a and I_a are similar. The lower PXRD resolution in the case of I_a may reflect the smaller particle size and crystal weathering over time. In order to conform the structure of I_a , a single crystal X-ray diffraction analysis has been carried out after a sample of C_a was converted to I_a with the sample being kept in the air for 24 h or in vacuum for 1 hour. Essentially the same unit cell was obtained. However, longer treatment times in the air or in vacuum was found to lead to crystal weathering. Further characterization of C_a and I_a came from ^1H NMR spectral studies. The results provided support for the conclusion that the molar ratio of CH_3CN and THF decreased on passing from C_a to I_a while all those of other components remained the same. The increasingly empty nature of the lattice as C_a is converted to I_a serves to enhance the molecular vibrations of perylene leading to a quenching of the

fluorescence intensity. Taken in concert, these results lead us to suggest that the main precursors of I_α retain a structure similar to C_α even after the loss of solvent components. A detail description of this chemistry has been added in the ESI before Supplementary Figure 16 as below:

As noted in the main text, when C_α was allowed to sit on the bench or in the air (298 K) for six days or subject to vacuum (2.0 kPa) for 5 hours conversion to I_α occurs. PXRD analyses revealed the crystalline structures of C_α and I_α are similar (Supplementary Figs. 16 and 17). The lower resolution seen for the PXRD spectrum of I_α may reflect the smaller particle size as well as time-dependent crystal weathering. Single crystal X-ray diffraction analyses of samples of C_α monitored under conditions used to produce I_α revealed that the sample retained essentially the same unit cell parameters when allowed sit for 24 h in the air or for in 1 hour in vacuum. However, Longer treatment times under both conditions led to further crystal weathering. Moreover, an ^1H NMR spectral study of these samples revealed a reduction in the relative molar ratio of CH_3CN and THF as a function of time (Supplementary Figs. 18 and 19). Taken in concert, these findings lead us to suggest that the intermediates leading to I_α maintain a structural framework similar to C_α even as some solvent components are lost. Subjecting C_α or I_α to grinding led to an amorphous material (A_m) as established by a PXRD study (Supplementary Fig. 16b). ^1H NMR spectral analysis revealed that A_m only contains $D_{4d}\text{-CDMB-8}$ and Py and is free of organic solvents (Supplementary Fig. 20). Treating A_m with THF/ CH_3CN (1/1, v/v) vapour could be used to access C_α (Supplementary Fig. 16c).

Supplementary Figure 16. a, Transformation between crystalline materials C_α , A_m , and I_α . b, Experimental PXRD patterns of C_α , I_α and A_m . c, PXRD studies of the reversible transformation between co-crystalline C_α and I_α and amorphous A_m .

Comment 3. In Supplementary Figure 17, the PXRD patterns of $C\alpha$ upon letting stand in air for 6 d and subjecting to vacuum for 5 h seem different to some extent, so I think these two solid-state forms could be respectively named as $I\alpha 1$ and $I\alpha 2$.

Answer: we thank the reviewer for the suggestion. We have named the two solid-state forms as $I\alpha 1$ and $I\alpha 2$ in Supplementary Figure 17.

Supplementary Figure 17. Time-dependent experimental PXRD patterns corresponding to the conversion of $C\alpha$ into $I\alpha 1$ seen upon letting stand in the air (a) or $I\alpha 2$ obtained by subjecting to vacuum (2.0 kPa) (b) at 298 K.

Comment 4. For the part of NB induced transformation, the intermediate crystalline solid $I\beta$ was described as the results of NB vapor covering the surface of $C\alpha$. As shown in PXRD patterns, the crystal structure of $C\alpha$ well remained after adsorption of NB for 20 min. Why not choose an intermediate state of structural transformation (it may be at about 1h according to time-dependent emission spectra of $C\alpha$ to $M\alpha$) where the PXRD started to change?

Answer: Yes; the PXRD of $C\alpha$ started to change after expose in NB vapour for 1 h according to the time-dependent PXRD spectra analyses, even if little change in the emission spectra is observable from 16 min to 1.5 h (Supplementary Figure 30. a₁-a₃). We choose crystalline solid $I\beta$ ($C\alpha$ after adsorption of NB for 20 min) as the intermediate state, because the structure of $I\beta$ (cf. Supplementary Figure 23a) was supported by the single crystal X-ray diffraction, PXRD, ^1H NMR spectral analyses. However, we agree that the solid obtained when $C\alpha$ is exposed to NB vapour for 1 h is likely a mixture of $I\beta$ and $M\alpha$. Since this form is poorly characterized, we do not discuss it in detail.

Figure R1. Time-dependent experimental PXRD patterns corresponding to the conversion of C_{α} into M_{α} seen upon letting expose in the nitrobenzene vapour.

Comment 5. In page 10, the first paragraph: “This matches the AIE emission produced by crystalline DPy .” Perylene doesn’t have AIE properties. The emission at 590nm was from aggregates of Py , which was more red-shifted and had much lower quantum yields compared to monomers and excimers. This was a classical manifestation of the ACQ phenomenon.

Answer: We thank the reviewer for noticing. We agree that the emission at 590 nm is reflects an ACQ phenomenon (for perylene) and have corrected the description in the manuscript as follows:

A change in the fluorescence colour from green to orange was seen when C_{α} was converted to a mixed crystalline form (e.g., M_{α} or M_{β}) by exposure to various organic solvents (either as vapours or used as for recrystallization) (cf. Supplementary Video B, and Supplementary Figs. 28, 34, and 46). The corresponding emission band appears at around 590 nm under conditions of UV illumination ($\lambda_{ex} = 365$ nm). This matches the **emission produced by the presumably aggregated Py in crystalline DPy** . This correspondence was taken as evidence that mixed crystalline solids, such as M_{α} or M_{β} , contain DPy crystalline domains and that these are responsible for the observed emission features. I_{α} and amorphous material (A_m) was characterized by a green-yellow emission around 500 nm (Fig. 3b, Supplementary Fig. 22). Crystals C_{α} and C_{δ} gave rise to a green luminescence ($\lambda_{em} = ca. 480$ nm) when subject to excitation at 365 nm using a handheld ultraviolet lamp. This emission is similar to that produced by perylene in solution at high concentrations (e.g., 10 mM in THF/ CH_3CN (1/1, v/v)) (Supplementary Fig. 21). This similarity leads us to propose that in the co-crystalline solid states, D_{4d} -**CDMB-8** serves to disperse the perylene molecules thus maintaining them in monomeric form. The fluorescence emission peak (λ_{em}), quantum yields (Φ_f),

and fluorescent lifetimes (τ_f) of C_α , A_m , M_α , M_β , D_{Py} and C_δ , are summarised (see Fig. 3 and Supplementary Fig. 44).

Comment 6. In Supplementary Figure 44 (a), the first line of the form which referred to the nonluminous crystalline materials should be “ C_β or C_γ ”.

Answer: We thank the referee for catching this. We have corrected the mistake in Supplementary Figure 44 (a) as below:

Supplementary Figure 44. Photophysical properties and normalized emission spectra of various solid forms considered in this study, namely C_α , C_β , C_γ , C_δ , I_α , A_m , I_β , M_α , M_β , and D_{Py} . **a**, Fluorescence images, emission peak (λ_{em}), quantum yields (Φ_f), and fluorescent lifetime (τ_f) determined using excitation at 365 nm. **b**, Normalized emission spectra ($\lambda_{em} = 365 \text{ nm}$).

Reviewer #2 (Remarks to the Author):

Comment 1. The volume ratio of solvents THF/ CH_3CN was 1/1, and the vapor was used assumed to be saturated in air at 298 K. So, did the author calculate the concentration ratio of the saturated vapor? If the ratio differs from that of the liquid solvents, which one is better for the molecular motion?

Answer: We thank the reviewer for the suggestion. Before choosing THF/ CH_3CN (1/1, v/v) as the solvent vapour system, a series of mixed THF/ CH_3CN solvent systems and their corresponding vapour components (near saturated in air at 298 K) were calculated (Table 1). We also studied their ability to promote the molecular motion needed to convert host-guest complex A_m to C_α . It was found most solvent mixtures (except THF/ CH_3CN (85/15, v/v)) would promote this transformation within 180 s (cf. table R1). For THF/ CH_3CN ratios from ca. 69:31 through to 36:64, the transformation rates proved similar and were faster than others cases tested (Figure, R2). Given this,

we chose a 1/1, v/v THF/CH₃CN ratio as the solvent system for the present initial communication. We do agree that further studies would be informative, but they lie outside the scope of the present study.

The saturated vapour pressures of THF and CH₃CN in air at 298 K are 637 kPa and 1118 kPa, respectively^[1]. Based on the following equation^[2]:

[1]. Speight, J. G. *Lange's Handbook of Chemistry, Sixteenth Edition* (MCGRAW-HILL, New York, 2005).

[2]. Atkins, P., Paula, J. D. W. H. *Atkins' Physical chemistry* (Freeman and Company, New York, 2006).

$$\frac{y_A}{y_B} = \frac{P_A^*}{P_B^*} \cdot \frac{x_A}{x_B}$$

Where P_A^* and P_B^* are saturated vapour pressures of THF and CH₃CN in air at 298 K,

respectively. $\frac{y_A}{y_B}$ is the concentration ratio of THF/CH₃CN in vapour, $\frac{x_A}{x_B}$ is the

volume ratio of solvents THF/CH₃CN.

Table R1. The concentration ratio of THF/CH₃CN in a given vapour mixture (y_A/y_B) and corresponding volume ratio of the liquid containing THF and CH₃CN (x_A/x_B)

y_A/y_B	x_A/x_B
1:10	85:15
1:4	69:31
1:3	63:37
1:2	53:47
1:1	36:64
1:2	22:78
1:3	16:84
1:4	12:88
1:10	05:95

Figure R2. **a**, Single crystal structures and photographs under a commercial ultraviolet lamp ($\lambda_{\text{ex}} = 365 \text{ nm}$) showing the transformation from amorphous \mathbf{A}_m with fluorescence green yellow colour to co-crystalline material \mathbf{C}_α with a characteristic fluorescence blue colour. **b**, Photographs of the THF/CH₃CN vapour-induced fluorescent colour changes for \mathbf{C}_α at different solvent mixture ratios, as seen at different time scales with visualization provided by a commercial ultraviolet lamp ($\lambda_{\text{ex}} = 365 \text{ nm}$).

Comment 2. Could the transform procedure accelerate by raising the temperature when induced by solvent vapors?

Answer: We thank the reviewer for this suggestion. As an initial temperature dependent study, we analyzed the transformation of \mathbf{C}_α to \mathbf{M}_β in the presence of THF vapour at 298 K, 318 K, and 338 K, respectively. This transformation could be accelerated by raising the temperature and as the saturated vapour pressure of the THF increased (Figure, R3). Again, further study of these effects would be interesting and we are pursuing such analyses in anticipation of writing up a full paper on this topic in due course.

Figure R3. **a**, Single crystal structures and photographs under a commercial ultraviolet lamp ($\lambda_{\text{ex}} = 365 \text{ nm}$) showing the transformation of the co-crystalline material C_{α} with fluorescence blue colour to a mixed crystalline species M_{β} with a characteristic fluorescence orange colour. **b**, Photographs of the THF vapour-induced fluorescent colour changes for C_{α} at different temperature, as seen at different time scales with visualization provided by a commercial ultraviolet lamp ($\lambda_{\text{ex}} = 365 \text{ nm}$).

Comment 3. The directly transformation between C_{δ} and A_m , C_{δ} and M_{β} , C_{δ} and M_{α} , M_{α} and M_{β} is not mentioned in the manuscript. Could they happen without state C_{α} ?

Answer: Yes; all these materials could be directly interconverted between one another; state C_{α} is not necessary for the transformations. Details of the interplay between C_{α} , C_{δ} , I_{β} , M_{α} , M_{β} , and A_m were studied by fluorescent spectrometry (Supplementary Figure 42, 43) and summarized in Supplementary Figure 41, including the transformation between C_{δ} and A_m , C_{δ} and M_{β} , C_{δ} and M_{α} . The transformation between M_{α} and M_{β} gives rise to no obvious change in emission spectra; however, qualitative ^1H NMR studies revealed they also could also be transformed directly between one another by NB or THF vapour treatment (Figure R4 and R5).

Supplementary Figure 41. Summary of the reversible transformations between C_α , C_δ , I_β , M_α , M_β , and A_m via grinding or treatment with the vapour forms of various organic solvents. Vapour I: A_m to C_α , THF/CH₃CN (1/1, v/v) (3 min), A_m to C_δ , Tol (40 s). vapour II: A_m to I_β , NB/THF/CH₃CN (1/1/1, v/v/v) (1 min). vapour III: A_m to M_β , THF (1 min), A_m to M_α , NB (4 h). vapour IV: C_α to I_β , PhNO₂ (20 min), C_δ to I_β , NB/THF/CH₃CN (1/1/1, v/v/v) (2 h). vapour V: I_β to C_α , THF/CH₃CN (1/1, v/v) (24 h), I_β to C_δ , Tol (8 h). vapour VI: C_α to M_β , THF (3 min), C_α to M_α , NB (21 h), C_δ to M_β , THF (10 min). vapour VII: M_β to C_α , THF/CH₃CN (1/1, v/v) (30 min), M_α to C_α , THF/CH₃CN (1/1, v/v) (24 h), M_β to C_δ , Tol (2 h), M_α to C_δ , Tol (12 h). vapour VIII: I_β to M_β , THF (5 min), I_β to M_α , NB (21 h). vapour IX: M_β to I_β , NB/THF/CH₃CN (1/1/1, v/v/v) (4 h), M_α to I_β , THF/CH₃CN (1/1, v/v) (1 min).

Figure R4. Single crystal structures and time-dependent expanded ¹H NMR (600 MHz) spectra (CDCl₃, 298 K, 1 mg/ml) showing the transformation from M_α to M_β by expose in THF vapour.

Figure R5. Single crystal structures and time-dependent expanded ¹H NMR (400 MHz) spectra (CDCl₃, 298 K, 1 mg/ml) showing the transformation from M_β to M_α by exposure in NB vapour.

Comment 4. Some English expressions need to be corrected:

The fifth line of the second paragraph: 'could serves' should be 'could serve';

The penultimate line of the sixteenth paragraph: 'a ultraviolet lamp' should be 'an ultraviolet lamp'.

Answer: We appreciate the reviewer's careful eye. These typographical errors have been corrected. The manuscript has also been reviewed carefully in an effort to minimize and hopefully eliminate all other errors in English usage.

REVIEWERS' COMMENTS:

Reviewer #1 (Remarks to the Author):

Publish as it is now.

Reviewer #2 (Remarks to the Author):

The revised manuscript could be accepted since all the revision points have been well addressed.